# Neural nets for modelling of scenarios and control of epidemics

Neverov A.V., Krivorotko O.I.
AI technologies for mathematical modeling of biological,
socio-economic and ecological processes
Sobolev Institute of mathematics SB RAS
Novosibirsk, Russia
a.v.neverov@math.nsc.ru
o.i.krivorotko@math.nsc.ru

## Abstract

The work considers the application of physically informed neural nets to the numerical analysis of epidemics, the effects of the introduction and ending of restrictive measures in SIR models and control in the form of the Hamilton-Jacobi-Bellman (HJB) equation.

Control problems for ordinary differential equations describe epidemic, social and other physics processes. We propose several scenarios of possible epidemic outcomes and formulate the optimal control problem. The usage of physically informed neural nets substantially decreases necessary man-hours to present characteristics of a differential equation and total computational time if according hardware (GPUs) is available. The proposed deep learning algorithm is compared to classical collocation approach. It is used in the most numerically challenging part of the problem, the nonlinear HJB partial differential equation, while classical methods are applied to the main ODE part of SIR-based model. Numerical results show a substantial effect of induced control in the decrease of the total number of infected and the time of active spread of an epidemic.

## 1 Introduction

The recent COVID-19 pandemic has shown the need to take into account socio-economic processes for a more accurate assessment of the effectiveness of anti-epidemic programs in a particular region Krivorotko & Kabanikhin (2024). One of the ways to model the epidemic process in the case of insufficiently detailed data (distribution of the population by region, socio-economic characteristics of the population, asymptomatic infected, distribution of the vaccinated population, etc.) is to build a generalized model with elements of optimal control.

We consider mean field game and mean field control approaches and investigate different ways of tackling control or setting players cost functional in modified SIR epidemiological model Kermack & McKendrick (1927).

Mean field games is a relatively novel approach to modeling large-scale systems with separate similar players with similar goals (cost functional). The central idea of this approach is to approximate the system by letting the number of players go to infinity, as soon as the number of players is already sufficiently large, which is a common situation for epidemiological problems. This approach allows turning numerous equations for discrete players into one Kolmogorov-Fokker-Planck equation for players distribution and Hamilton-Jacobi-Bellman equation for their control; hence, we may model the behavior of people based on their own decisions of optimal control, but with fewer equations to solve. It was originally proposed by Lasry, Lions in Lasry & Lions (2007). The analytical solution is known only for specific form of problem setup that leads to Ricatti equations (Arabneydi et al., 2020). Existence and uniqueness of solution is proven under special conditions of monotonicity (Achdou et al., 2016).

Numerical solution of the mentioned above pair of equations is complicated with the fact that the KFP equation is forward in time and HJB is backward. There are several finite difference schemes that utilize an iterative approach to sequential solution of one equation with fixed solution of the other. The methods proposed in this work implement an approach with iterations of non-linearity only, with simultaneous solutions of both equations.

Machine learning approaches (such as auto-regressions, various neural nets, support vector machines and other specialized tools (Kontopoulou et al., 2023)) appeared to have more flexible descriptive properties, than classical models, based on differential equations. However, those methods may not hold fundamental laws, such as conservation laws. Another obstacle to machine learning models usage is their lack of interpretability, that does not allow for building recommendation systems or making conclusions on the causality of phenomena. (Yi et al., 2022; Feng et al., 2021).

Neural nets obtain one of the best approximation proberties (Hornik et al., 1989) and may be used as simple extrapolator of time series, however they may be combined with classical approaches in the way of PINN, that utilizes partial differential model as additional source of information for approximation. There is already done research in implementation of PINN for SIR-like models (Bertaglia et al., 2022; Nguyen et al., 2024).

We propose combination of mentioned above techniques to combine advantages of flexibility of PINN with interpretability of classical SIR model and complexity and details of model of mean field game approach.

The paper has the following structure: in section 2 we present introduction to general mean field game problem setup; in section 3 we propose epidemiological mean field game model with players controlling parameters of classical SIR model and system of equations associated with it; in section 4 we describe numerical methods used to solve resulting system of equations including PINN; the numerical results are analyzed in section 5.

## 2  Mean field game problem formulation

Mean field type models arise in problems of modeling a large number of similar objects that are commonly particles. To construct mean field game models it is assumed that all objects are identical players, whose state is described with a stochastic differential equation in the field $\mathcal{T} = [0, T], T > 0, \Omega \subset R^n, n \in N$:

$$\begin{cases} dX(t) = g(t, X(t), \alpha)dt + \sigma dw(t), & t \in \mathcal{T}, \\ X(0) = X_0, \end{cases} \tag{1}$$

where $g : \mathcal{T} \times \Omega \times A \to R^n$ is a players state control function,
$\alpha \in A$ is player's strategy, $A := \{\alpha : \mathcal{T} \times \Omega \to R\}$,
$\sigma = \text{const}$ is a dispersion of random walk of agents,
$dw$ is a standard Wiener process increment.

Every players goal is to minimize its own cost functional (Achdou et al., 2020a)

$$J(X, \alpha, m) = \mathbf{E}\left[ \int\limits_0^T f(t, X(t), m, \alpha)dt + h(X(T), m(T)) \right]. \tag{2}$$

Where $m : M = \mathcal{T} \times \Omega \to R^+$, $\int_\Omega m(t, x)dx = 1$ is a distribution function of all players;
$f : \mathcal{T} \times \Omega \times M \times A \to R$ is a running cost of a game; $h : \Omega \times (\Omega \to R^+) \to R$ is a terminal cost of a game.

As the number of players is very large, it can be approximated with an infinite number of players and transferred from the SDE (1) to the Kolmogorov-Fokker-Plank (KFP) equation:

$$\begin{cases} \dfrac{\partial m}{\partial t}(t, x) = \dfrac{1}{2}\nabla^2(\sigma m(t, x)) + \nabla \cdot (g(t, x, \alpha)m(t, x)), & t \in \mathcal{T}, \\ m(0, x) = m_0(x), & x \in \Omega. \end{cases} \tag{3}$$

Thus we move from evaluating influence of every discrete player on every other, to evaluating influence of mean distribution of players on one test player.

To obtain the optimal conditions for formulated mean field game problem introduce two functions that are set for modified problem of minimization of (2) with start at time $t$ and $X_{x_t}(t) = x_t = \text{const}$:

$$\hat{J}(t, x_t, \alpha) = \mathbf{E}\left[\int_t^T f(s, X_{x_t}(s), m, \alpha)ds + h(X(T), m(T))\right],$$

$$\psi(t, x_t) = \inf_{\alpha \in A} \hat{J}(t, x_t, \alpha), \qquad \psi : A.$$

Using the dynamical programming principle, we get

$$\psi(t, x_t) = \inf_{\alpha \in A} \mathbf{E}\left[\int_t^{t+\tau} f(s, X_{x_t}(s), m, \alpha)ds\right] + \psi(t + \tau, X_{x_t}(t + \tau)). \tag{4}$$

Now with use of the Ito formula for $\psi(t, x)$:

$$\psi(t + \tau, X_{x_t}(t + \tau)) = \psi(t, x_t) + \mathbf{E}\int_t^{t+\tau}\left[\frac{\partial \psi}{\partial t} + g(s, X_{x_t}(s), \alpha)\nabla\psi(s, X_{x_t}(s)) + \right.$$
$$\left. + \frac{1}{2}\sigma\nabla^2\psi(s, X_{x_t}(s))\right]ds + o(dw_s). \tag{5}$$

Substitution (5) into (4) results in

$$0 = \inf_{\alpha \in A} \mathbf{E}\left[\int_t^{t+\tau} f(s, X_{x_t}(s), m, \alpha)ds + \frac{\partial \psi}{\partial t} + \right.$$
$$\left. + g(s, X_{x_t}(s), \alpha)\nabla\psi(s, X_{x_t}(s)) + \frac{1}{2}\sigma\nabla^2\psi(s, X_{x_t}(s))ds\right]. \tag{6}$$

Dividing (6) by $\tau$ and tending $\tau$ to 0, we get the optimal condition in the form of Hamilton-Jacobi-Bellman equation (HJB):

$$0 = \frac{\partial \psi}{\partial t} + \frac{1}{2}\sigma\nabla^2\psi + \inf_{\alpha}\left[g(t, x, \alpha)\nabla\psi + f(t, x, m, \alpha)\right] \tag{7}$$

and, in general case, the nonlinear equation that allows to evaluate optimal control $\alpha$ from solution of HJB (7):

$$\frac{\partial g}{\partial \alpha}\nabla\psi + \frac{\partial f}{\partial \alpha} = 0. \tag{8}$$

From the definition of $\psi$ we get the terminal condition for HJB equation:

$$\psi(T, x_t) = h(X(T), m(T)). \tag{9}$$

Thus, in mean field games formulation we get that the optimal problem is described by three equations in the form (3), (7), (8).

Summing up, we get the complete problem of MFG in the form:

$$\begin{cases} \dfrac{\partial \psi}{\partial t} + \dfrac{1}{2}\sigma\nabla^2\psi + H(\nabla\psi) = 0, & x \in \Omega, \\ \psi(T, x) = 0, & t \in \mathcal{T}, \\ \dfrac{\partial m}{\partial t} - \dfrac{1}{2}\sigma\nabla^2 m + \nabla \cdot [H(\nabla\psi)m] = 0, \\ m(0, x) = m_0(x). \end{cases} \tag{10}$$

where Hamiltonian $H(\nabla \psi) = \inf\limits_{\alpha \in A} \left[ g(t, x, \alpha) \nabla \psi(t, x) + f(t, x, m, \alpha) \right]$.

## 3 Epidemiological mean field game

The mean field approach is widely used for modeling and control analysis in epidemiology, economics, social processes and other natural fields. The recent research shows that model construction and investigation of control programs for COVID-19 epidemic are analyzed for different regions, strains and restriction measures Tembine (2020); Petrakova & Krivorotko (2022); Roy et al. (2023). The coefficients and initial conditions of mean field models are often estimated from general statistics, which leads to inaccurate control results. Identification of mean field model parameters is labor-intensive, its studies are given in the paper Ding et al. (2022); Chow et al. (2022); Liu et al. (2023); Petrakova (2024). It is shown that under certain conditions on the parameters and control function, sensitive parameters of the models can be restored.

Further in this section, we formulate a mean field model based on the basic SIR model in order to demonstrate a new algorithm for numerically solving this problem.

### 3.1 SIR model

The basic epidemiological model based on ordinary differential equations was initially presented in Kermack & McKendrick (1927). It is describing circulation of single disease in a closed population and is based on the mass balance law. The model is described with the following system of ODEs in the domain $t \in \mathcal{T}$:

$$\begin{cases} \dfrac{dS}{dt} = -\beta S(t)I(t), & S(0) = 1 - I_0, \\ \dfrac{dI}{dt} = \beta S(t)I(t) - \gamma I(t), & I(0) = I_0, \\ \dfrac{dR}{dt} = \gamma I(t), & R(0) = 0. \end{cases} \tag{11}$$

Every individual in population may be in one of three compartments: Susceptible, Infected and Removed and total volume of people stays constant $1 = S(t) + I(t) + R(t)$, $S, I, R : \mathcal{T} \to R^+$.

### 3.2 SIR-MFG model

With beginning of an epidemic we consider people to have different attitude toward self isolation, wearing masks and other measures of infection prevention and treatment measures may differ due to economical status of person. Thus, we propose models, that treats people as players and implements player control of infectivity and recovery rate. The goal of every player is fixed similar for every player, that is to minimize their own losses throughout the modeling time.

We introduce control of the infectivity rate of every player in susceptible compartment.

$$\beta : \Omega \to R^+, \quad S(t) = \int\limits_{\Omega} s(t, x)dx.$$

The process infectivity rate change is described with stochastic equation (1).

That leads to KFP equation on density of players $s$:

$$\begin{cases} \dfrac{ds}{dt}(t, x) = -\beta(x)s(t, x)I(t) + \frac{\sigma}{2}\Delta s - \nabla \cdot (g(\alpha)s) & t \in \mathcal{T}, \\ s(0, x) = s_0(x) & x \in \Omega. \end{cases}$$

Then we introduce cost functional for players that they are to minimize:

$$\inf_{\alpha \in A} \mathcal{J} = \mathbf{E} \int_0^T \left[ w_1 I^2(t) + w_2 X(t)I(t) + w_3(1 - X(t))^2 + w_0 g^2(t, X(t), \alpha) \right] dt, \qquad (12)$$

where $w_1, w_2, w_3, w_0 \in R^+$ are weights, $I^2$ is a cost of infected people being not able to work, $X(t)I(t)$ is a cost of risk of being infected, $(1 - X(t))^2$ is a cost of self-isolation and other measures, $g^2$ is cost of adaptation to changes in those measures.

Then the HJB equation is obtained in the form:

$$\begin{cases} \dfrac{d\psi}{dt} = \dfrac{\sigma}{2}\Delta\psi - \left[ w_1 I^2(t) + w_2 xI(t) + w_3(1 - x)^2 + \dfrac{1}{2w_0}(\nabla\psi)^2 \right] & t \in \mathcal{T}, \\ \psi(T, x) = 0 & x \in \Omega, \end{cases}$$

and the optimal control according to equation 8 is as follows:

$$\alpha(t, x) = -\frac{\nabla\psi(t, x)}{2w_0}.$$

The resulting MFG system in domain derived with modification of system (11) has the following form:

$$\begin{cases} \dfrac{\partial s}{\partial t} = -\beta(x)sI(t) + \dfrac{\sigma}{2}\Delta s - \dfrac{1}{2w_0}\nabla \cdot (\nabla\psi s), & s(0, x) = s_0(x), \\ \dfrac{\partial \psi}{\partial t} = \dfrac{\sigma}{2}\Delta\psi - \left[ w_1 I^2(t) + w_2 xI(t) + w_3(1 - x)^2 + \dfrac{1}{2w_0}(\nabla\psi)^2 \right], & \psi(t, x) = 0, \\ \dfrac{dI}{dt} = \int_\Omega \beta(x)s(\cdot, x)I dx - \gamma I, & I(0) = I_0, \\ \dfrac{dR}{dt} = \gamma I, & R(0) = 0. \end{cases} \qquad (13)$$

Alternative modification for control of $\gamma$:

$$\gamma : \Omega \to R^+, \quad I(t) = \int_\Omega i(t, x)dx.$$

The process removal rate change is described with the stochastic equation identical to the (1)

$$\begin{cases} \dfrac{dS}{dt} = -\beta S(t)I(t), & S(0) = N - I_0, \\ \dfrac{di}{dt} = i(\beta S(t) - \gamma(x)) + \dfrac{\sigma}{2}\Delta i - \dfrac{1}{2w_0}\nabla \cdot (\nabla\psi i), & i(0, x) = i_0(x), \\ \dfrac{d\psi}{dt} = \dfrac{\sigma}{2}\Delta\psi - \left[ w_1 I^2(t) + w_2 xI(t) + w_3(1 - x)^2 + \dfrac{1}{2w_0}(\nabla\psi)^2 \right], & \psi(t, x) = 0, \\ \dfrac{dR}{dt} = \int_\Omega i(\cdot, x)\gamma(x)dx, & R(0) = 0. \end{cases} \qquad (14)$$

## 4 Numerical approaches

In all numerical experiments below we consider:

$$\begin{aligned} \gamma(x) = \beta(x) &= x, \\ g(t, x, \alpha) &= \alpha(t, x), \quad x \in \Omega = [0, 1], \\ T = 1 &\implies t \in [0, 1]. \end{aligned}$$

## 4.1 Finite difference method (FDM)

Basic finite difference schemes for solution of MFG system of equations were presented in works Achdou & Capuzzo-Dolcetta (2010); Achdou et al. (2012); Achdou (2013). Moreover in Achdou et al. (2020b) several iterative approaches to solution nonlinear, forward-backward are discussed.

We implement simple implicit numerical scheme with 3-point Laplassian and Newton linearisation of non-linear part:

In work Lachapelle et al. (2010) the monotone scheme is described, however it lacks simultaneous update of KFP equation sulution and control, thus this method does not guarantee convergense to local optimum (global in case of convex $J$), according to Salomon & Turinici (2011). Thus we implement scheme from Lachapelle et al. (2010) with straightforward approach of solving at all time points simultaneously to achieve simultaneous update of KFP solution and control.

## 4.2 Collocations method

Collocations method utilizes representation of solution in a chosen basis as an approximation of exact solution (Shapeev et al., 2021; Belyaev et al., 2022). It allows us to substitute it into the PDEs and explicitly differentiate basis functions. By evaluating resulting functions in chosen collocation points, we may obtain equations for unknown basis coefficients of solution.

For problem in form: $A(t,x)u(t,x) = f(t,x)$ with solution searched in form $u(t,x) = \sum a_k \varphi_k(t,x)$ collocation method may be written in form (1). Where $N_x, N_t, N_\varphi$ are amounts of spatial steps, time steps and basis functions respectively, $\{\varphi_i(t,x)\}$ is a set of basis functions, $P^c, P^b, P^i$ are sets of collocation, border and initial condition points.

As long as our problem is nonlinear, we use Newton linearisation and apply this algorithm several times, while updating solution approximation on every iteration.

It worth noting that systems (13) and (14) include integral parts. However, collocation methods are designed for local satisfaction of equations. This obstacle is overcame the following way, inspired by the following works: (Wang & Yang, 2024; Pang et al., 2020; Yuan et al., 2022).

We represent the integral the following way we formally add $x$ to functions independent of $x$ and add condition $\frac{\partial}{\partial x} \cdot = 0$:

$$
\begin{cases}
\beta_{sI}(t,x) = \int\limits_\Omega \beta(\xi) s(\xi,t) I(\xi,t) d\xi, & x \in \delta\Omega^+, \\
\dfrac{\partial \beta_{sI}}{\partial x}(t,x) = \beta s I(t,x), & x \in \Omega, \\
\beta_{sI}(t,x) = 0, & x \in \delta\Omega^-, \\
\dfrac{\partial I}{\partial t}(t,x) = \beta_{sI}(t,x) - \gamma I(t,x), & x \in \delta\Omega^+, \\
\dfrac{\partial R}{\partial t}(t,x) = \gamma I(t,x), & x \in \delta\Omega^+, \\
\dfrac{\partial I}{\partial x}(t,x) = \dfrac{\partial R}{\partial x}(t,x) = 0, & x \in \Omega, \, t \in \mathcal{T}.
\end{cases}
$$

In our case $\beta_{sI} : M$ and since $\Omega = [0,1]$, then $\delta\Omega^- = \{0\}$, $\delta\Omega^+ = \{1\}$.

For numerical experiments we choose $N_x$ and $P^c$ such that $N_x \cdot |P^c|$ was approximately equal to $N_x$ in FDM.

## 4.3 Physically informed neural nets

PINN (physically informed neural net) combines ideas of collocation method and neural net error optimisation with help of automatic differentiation. The same way, as in collocation

---

**Algorithm 1** Collocation method

---

**Require:** $N_x, N_t, N_\varphi, \{\varphi_i(t,x)\}_{i=\overline{1,N_\varphi}}, P^c, P^b, P^i$

    $h, \tau \leftarrow N_x^{-1}, N_t^{-1}$

    $c_{i,j} \leftarrow (h(i + 1/2), \tau(j + 1/2))$                            ▷ Grid cells centers

    **for** every $P$ in $\{P^c, P^b, P^i\}$ **do**

        **for** every $p$ in $P$ **do**

            $p = (p_0, p_1) \leftarrow p + c_{i,j}$

            **if** $p_0 \in 0, 1, P == P^b$ **then**               ▷ Point on border

                Get equation $\sum_{k=1}^{N_b} \varphi_k(p)a_k = u_b(p)$

            **else if** $p_1 == 0, P == P^c$ **then**           ▷ Point on line $t = 0$

                Get equation $\sum_{k=1}^{N_b} \varphi_k(p)a_k = u_i(p)$

            **else**                                     ▷ Collocation point

                Get equation $\sum_{k=1}^{N_b} A(p)\varphi_k(p)a_k = f(p)$

            **end if**

        **end for**

    **end for**

    Collect all obtained linear equations and solve as single system.

---

method above, the solution is approximated with neural net. It and its derivatives are evaluated in collocation points and substituted into PDEs and border conditions to obtain residual errors, that are used in back propagation algorithm. The similarity may be seen in comparison of algorithms 1, 2.

---

**Algorithm 2** PINN

---

**Require:** $N_x, N_t, P^c, P^b, P^i$

    Residual $R, \delta R, \leftarrow 0, \infty$

    **while** $\delta R > \varepsilon$ **do**

        $R_0 \leftarrow R$

        **for** every $P$ in $\{P^c, P^b, P^i\}$ **do**

            **for** every $p$ in $P$ **do**

                Evaluate $u^{NN}(p)$ and its derivatives $u_x^{NN}(p), u_x x^{NN}(p), u_t^{NN}(p)$

                **if** $P == P^b$ **then**

                    Add to residual $u^{NN}(p) - u_b(p)$

                **else if** $P == P^i$ **then**

                    Add to residual $u^{NN}(p) - u_o(p)$

                **else if** $P == P^c$ **then**

                    Add to residual $A(p)u^{NN}(p) - f(p)$

                **end if**

            **end for**

        **end for**

        Perform backward error propagation with residual obtained

        $\delta R \leftarrow |R - R_0|$

    **end while**

---

For numerical experiments we choose and $P^c$ such that $|P^c|$ was approximately equal to $N_x$ in FDM. The neural net was implemented with fully connected architecture with 20 hidden layers with width of 10 neurons per layer, as it shows sufficient approximation properties and learning speed.. The activation function was chosen to be *tanh*.

## 5 Numerical experiments

### 5.1 Problem examples

We present several examples of numerical solutions of problems (13), (14) we variate weights of the cost functionals. The baseline scenario is modeled with standard SIR model and the baseline controlled scenario has parameters $w_i = 1, i \in \{0, 1, 2, 3\}$.

To illustrate the influence of cost of being infected or staying isolated we variate weights $w_2$ and $w_3$.

### 5.2 Numerical results

Results are presented in the Table 1 and in Figures 1, 2, 3.

Table 1: Numerical experiments results

| Controlled variable | $w_2$ | $w_3$ | Achieved $R$ |
|---|---|---|---|
| $\beta$ | 1 | 1 | 3.4e-5 |
| | 1 | 10 | 6.4e-5 |
| | 5 | 10 | 1.9e-4 |
| $\gamma$ | 1 | 1 | 2.1e-4 |
| | 1 | 10 | 8.4e-4 |
| | 5 | 10 | 2.5e-3 |

All the numerical experiments took 3100 seconds and 50 000 epochs to converge.

Numerical experiments with collocation method converge, but only for smaller time periods T, that lead to different problem setup, that are not comparable with the results for the PINN. In Figure 4 we may see intermediate results for collocation method with grid 40 by 10 and polynomial basis of power 4. That leads to comparable amount of parameters of the approximation. The linearisation is still was needed and it took 84 iterations to converge, that took approximately 2600 seconds for the time period 2 times smaller, than the one for the PINN.

However, even for smaller time period, it did converge to a local residual minimum, that does not present adequate results (i.e. Infected group became negative).

## 6 Conclusion

The numerical results obtained show that in the trade-off of hospitalization and self-isolation costs, agents tend to minimize one of them and stay with this strategy until the end of the epidemic (modeling time), or stay with the starting strategy. It seems that as soon as epidemics comes to an end and number of infected declines, it would be profitable to change strategy to the opposite, however the small numbers of susceptible leads to small number of isolated population. Thus this transit happens later in time (after observed time period in experiments), or may happen with narrow set of parameters, that is close to situation with no change in strategy.

In all the scenarios presence of infection cost leads to decrease in simultaneous number of infected, that in practice could lead to earlier end of epidemic, but the nature of the model allows only situations with all the populations going through the epidemic process.

Code availability

The basis frameworks for the realisation of PINN and collocation methods are published at github.com/ANever/sib-pinn and github.com/ANever/cls.

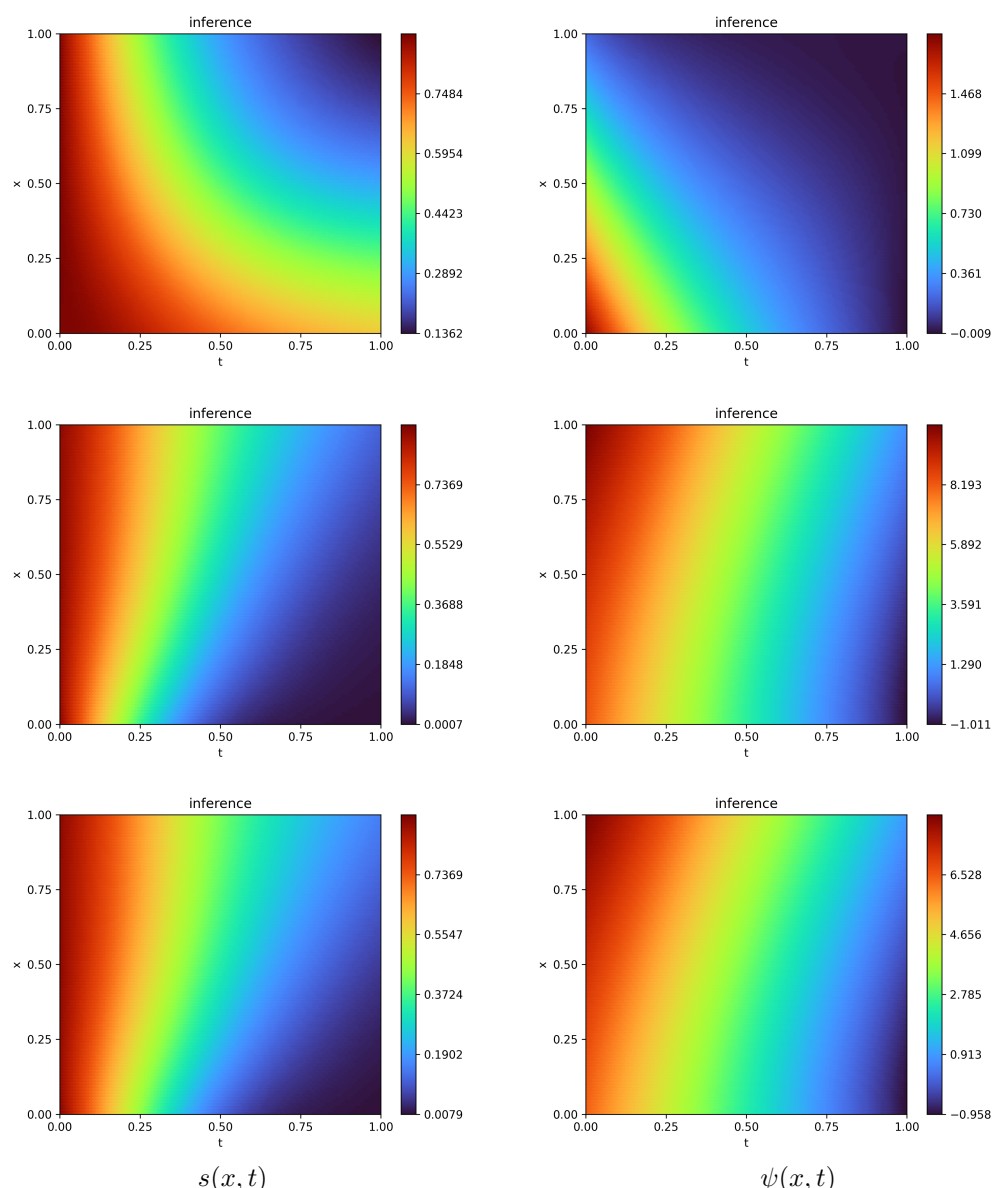

$$s(x,t) \qquad\qquad \psi(x,t)$$

Figure 1: Numerical results of modeling problem 13 for pairs of parameters $(w_2, w_3)$ (top to bottom): (1,1), (1,10), (5,10).

Acknowledgments

This work is supported by the Government research assignment for the Sobolev Institute of Mathematics SB RAS, project FWNF-2024-0002.

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

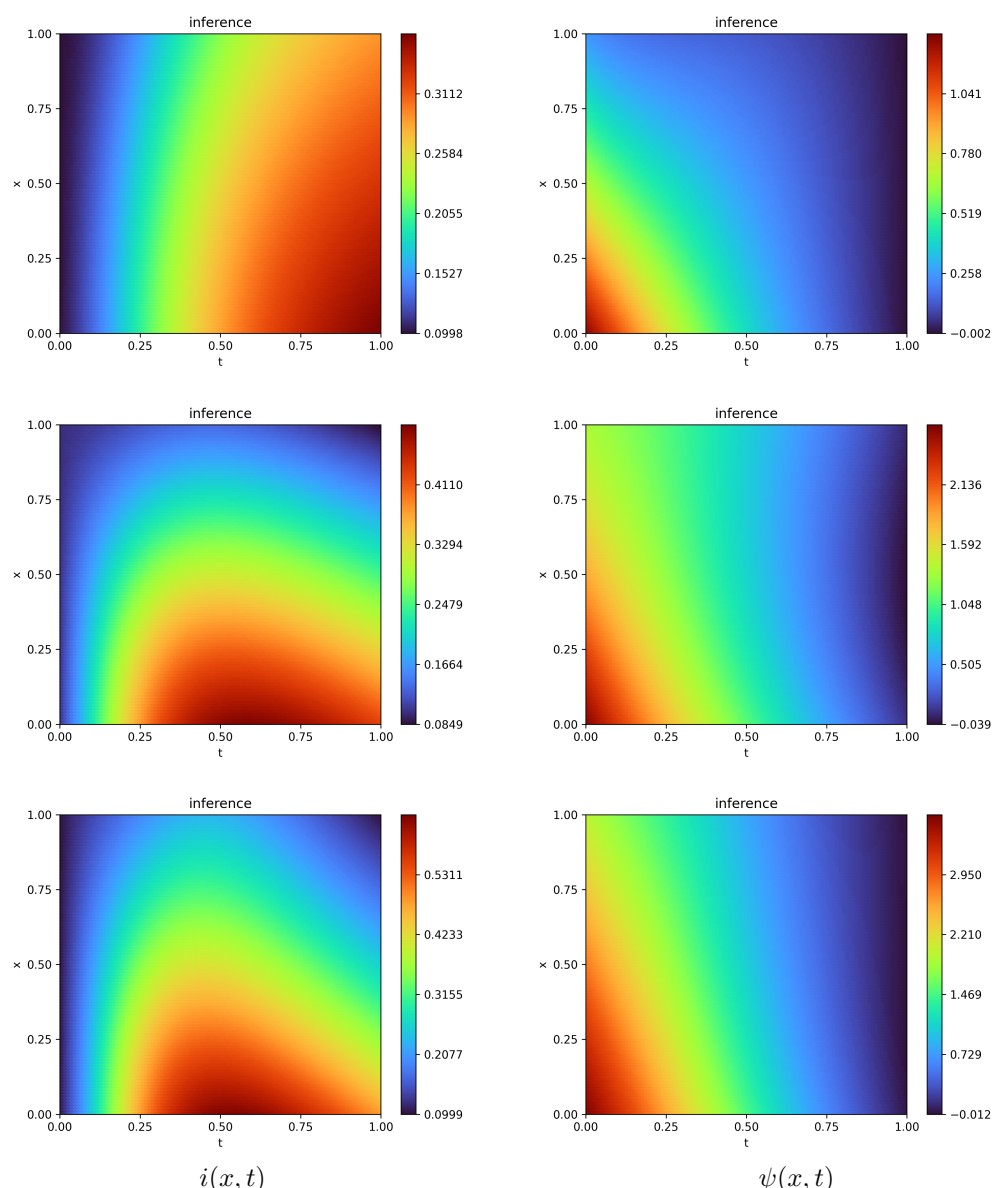

$$i(x, t) \qquad\qquad \psi(x, t)$$

Figure 2: Numerical results of modeling problem (14) for pairs of parameters $(w_2, w_3)$ (top to bottom): (1,1), (1,10), (5,10).

7170. doi: 10.1137/090758477.

Yves Achdou, Fabio Camilli, and Italo Capuzzo-Dolcetta. Mean Field Games: Numerical Methods for the Planning Problem. SIAM Journal on Control and Optimization, 50(1): 77–109, January 2012. ISSN 0363-0129, 1095-7138. doi: 10.1137/100790069.

Yves Achdou, Martino Bardi, and Marco Cirant. Mean Field Games models of segregation, July 2016.

Yves Achdou, Pierre Cardaliaguet, François Delarue, Alessio Porretta, and Filippo Santambrogio. Mean Field Games: Cetraro, Italy 2019, volume 2281 of Lecture Notes in Mathematics. Springer International Publishing, Cham, 2020a. ISBN 978-3-030-59836-5 978-3-030-59837-2. doi: 10.1007/978-3-030-59837-2.

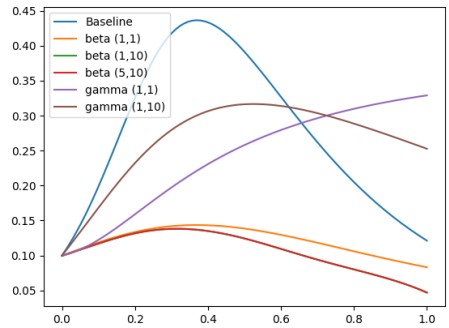

Figure 3: Numerical results for infected compartment in all numerical experiments from Table 1.

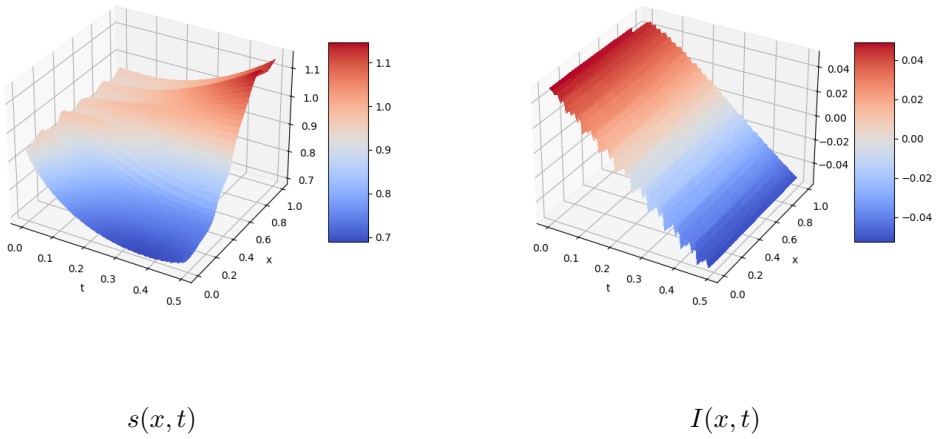

$s(x,t)$ $\qquad\qquad\qquad\qquad\qquad\qquad$ $I(x,t)$

Figure 4: Numerical results of modeling problem (14) for pairs of parameters $w_2 = w_3 = 1$ with collocation method.

Yves Achdou, Pierre Cardaliaguet, François Delarue, Alessio Porretta, and Filippo Santambrogio. Mean Field Games: Cetraro, Italy 2019, volume 2281 of Lecture Notes in Mathematics. Springer International Publishing, Cham, 2020b. ISBN 978-3-030-59836-5 978-3-030-59837-2. doi: 10.1007/978-3-030-59837-2.

Jalal Arabneydi, Amir G. Aghdam, and Roland P. Malhamé. Explicit Sequential Equilibria in LQ Deep Structured Games and Weighted Mean-Field Games: A Unified Non-Standard Riccati Equation, 2020. URL http://arxiv.org/abs/1912.03931.

V. A. Belyaev, L. S. Bryndin, S. K. Golushko, B. V. Semisalov, and V. P. Shapeev. h-, p-, and hp-versions of the least-squares collocation method for solving boundary value problems for biharmonic equation in irregular domains and their applications. Computational Mathematics and Mathematical Physics, 62(4):517–537, 2022. doi: 10.1134/S0965542522040029.

Giulia Bertaglia, Chuan Lu, Lorenzo Pareschi, and Xueyu Zhu. Asymptotic-preserving neural networks for multiscale hyperbolic models of epidemic spread. Mathemati-

cal Models and Methods in Applied Sciences, 32(10):1949–1985, 2022. doi: 10.1142/S0218202522500452. URL https://doi.org/10.1142/S0218202522500452.

Y.T. Chow, S.W. Fung, S. Liu, L. Nurbekyan, and S. Osher. A numerical algorithm for inverse problem from partial boundary measurement arising from mean field game problem. Inverse Problems, 39:014001, 2022.

Lisang Ding, Wuchen Li, Stanley Osher, and Wotao Yin. A mean field game inverse problem. J Sci Comput, 92:7, 2022. doi: 10.1007/s10915-022-01825-8.

S. Feng, Z. Feng, C. Ling, C. Chang, and Z. Feng. Prediction of the covid-19 epidemic trends based on seir and ai models. PLoS ONE, 16(1):e0245101, 2021.

Kurt Hornik, Maxwell Stinchcombe, and Halbert White. Multilayer feedforward networks are universal approximators. Neural Networks, 2(5):359–366, 1989. ISSN 0893-6080. doi: https://doi.org/10.1016/0893-6080(89)90020-8. URL https://www.sciencedirect.com/science/article/pii/0893608089900208.

O. W. Kermack and A. G. McKendrick. A contribution to the mathematical theory of epidemics. Proceedings of the Royal Society of London. Series A, Containing Papers of a Mathematical and Physical Character, 115(772):700–721, 1927. ISSN 0950-1207, 2053-9150. doi: 10.1098/rspa.1927.0118. URL https://royalsocietypublishing.org/doi/10.1098/rspa.1927.0118.

Vaia I. Kontopoulou, Athanasios D. Panagopoulos, Ioannis Kakkos, and George K. Matsopoulos. A review of arima vs. machine learning approaches for time series forecasting in data driven networks. Future Internet, 15(8), 2023. ISSN 1999-5903. doi: 10.3390/fi15080255. URL https://www.mdpi.com/1999-5903/15/8/255.

Olga Krivorotko and Sergey Kabanikhin. Artificial intelligence for covid-19 spread modeling. Journal of Inverse and Ill-posed Problems, 32(2):297–332, 2024. doi: doi: 10.1515/jiip-2024-0013. URL https://doi.org/10.1515/jiip-2024-0013.

Aime Lachapelle, Julien Salomon, and Gabriel Turinici. COMPUTATION OF MEAN FIELD EQUILIBRIA IN ECONOMICS. Mathematical Models and Methods in Applied Sciences, 20(04):567–588, April 2010. ISSN 0218-2025, 1793-6314. doi: 10.1142/S0218202510004349.

Jean-Michel Lasry and Pierre-Louis Lions. Mean field games. Japanese journal of mathematics, 2(1):229–260, 2007. ISSN 0289-2316. doi: 10.1007/s11537-007-0657-8.

Hongyu Liu, Chenchen Mou, and Shen Zhang. Inverse problems for mean field games. Inverse Problems, 39(8):085003, jun 2023. doi: 10.1088/1361-6420/acdd90. URL https://dx.doi.org/10.1088/1361-6420/acdd90.

Thang Nguyen, Dung Nguyen, Kha Pham, and Truyen Tran. Mp-pinn: A multi-phase physics-informed neural network for epidemic forecasting, 2024. URL https://arxiv.org/abs/2411.06781.

Guofei Pang, Marta D'Elia, Michael Parks, and George E. Karniadakis. npinns: nonlocal physics-informed neural networks for a parametrized nonlocal universal laplacian operator. algorithms and applications, 2020. URL https://arxiv.org/abs/2004.04276.

Viktoriya Petrakova. Inverse coefficient problem for epidemiological mean-field formulation. Mathematics, 12(22), 2024. doi: 10.3390/math12223581. URL https://www.mdpi.com/2227-7390/12/22/3581.

Viktoriya Petrakova and Olga Krivorotko. Mean field game for modeling of covid-19 spread. Journal of Mathematical Analysis and Applications, 514(1):126271, 2022. ISSN 0022-247X. doi: https://doi.org/10.1016/j.jmaa.2022.126271. URL https://www.sciencedirect.com/science/article/pii/S0022247X22002852.

A. Roy, C. Singh, and Y. Narahari. Recent advances in modeling and control of epidemics using a mean field approach. Sadhana, 48:207, 2023.

Julien Salomon and Gabriel Turinici. A monotonic method for nonlinear optimal control problems with concave dependence on the state. International Journal of Control, 84(3): 551–562, March 2011. ISSN 0020-7179, 1366-5820. doi: 10.1080/00207179.2011.562548.

Vasily Shapeev, Sergey Golushko, Vasily Belyaev, Luka Bryndin, and Pavel Kirillov. New versions of the least-squares collocation method for solving differential and integral equations. volume 1715, pp. 012031, 01 2021. doi: 10.1088/1742-6596/1715/1/012031.

Hamidou Tembine. Covid-19: Data-driven mean-field-type game perspective. Games, 11 (4), 2020. doi: 10.3390/g11040051. URL https://www.mdpi.com/2073-4336/11/4/51.

Yeping Wang and Shihao Yang. Coupled integral pinn for conservation law, 2024. URL https://arxiv.org/abs/2411.11276.

Junfei Yi, Hui Zhang, Jianxu Mao, Yurong Chen, Hang Zhong, and Yaonan Wang. Review on the covid-19 pandemic prevention and control system based on ai. Engineering Applications of Artificial Intelligence, 114:105184, 2022. ISSN 0952-1976. doi: https://doi.org/10.1016/j.engappai.2022.105184. URL https://www.sciencedirect.com/science/article/pii/S0952197622002858.

Lei Yuan, Yi-Qing Ni, Xiang-Yun Deng, and Shuo Hao. A-pinn: Auxiliary physics informed neural networks for forward and inverse problems of nonlinear integro-differential equations. Journal of Computational Physics, 462:111260, 2022. ISSN 0021-9991. doi: https://doi.org/10.1016/j.jcp.2022.111260. URL https://www.sciencedirect.com/science/article/pii/S0021999122003229.

