# OpenReview forum: "Neural nets for modelling of scenarios and control of epidemics"
_mathai.club/MathAI/2025/Conference — MathAI 2025 Oral_

### Official Review · Reviewer_Acc6 · 2025-02-26
**The paper addresses the challenging problem of modeling and controlling epidemic outbreaks using neural networks integrated with mathematical epidemiology. It proposes a physics-informed neural network approach (referred to as "physically informed neural nets" by the authors) to forecast epidemic scenarios and to compute optimal intervention strategies in an SIR (Susceptible-Infected-Removed) model​. The authors formulate the epidemic control task as a mean-field game (MFG) optimal control problem, where individuals (or “players”) choose strategies (like self-isolation or treatment efforts) that influence the infection spread. A system of coupled differential equations is derived – combining the classical SIR ODEs with a Hamilton-Jacobi-Bellman (HJB) partial differential equation – to represent the evolution of the epidemic and the optimal control policy. To solve this system, the paper introduces a hybrid numerical method: a deep learning algorithm based on physics-informed neural networks (PINNs) for the HJB PDE, combined with classical numerical schemes (collocation/finite-difference methods) for the SIR ODE components​. Numerical experiments are presented to illustrate various epidemic scenarios (e.g. different weights on costs of infection vs. isolation) and to demonstrate that the optimized control policies can significantly reduce the total infections and shorten the epidemic duration​. In summary, the paper’s contributions lie in formulating an epidemic optimal control as a mean-field game and in leveraging PINNs to efficiently solve the resulting complex system of equations for forecasting and control.**

**Rating:** 5
**Confidence:** 4

**Review:**

Strengths:
- The paper combines several advanced approaches – epidemiological SIR modeling, optimal control via mean-field game theory, and physics-informed neural networks – into a unified framework. This interdisciplinary fusion is relatively novel. By integrating a PINN with a classical collocation method, the authors exploit the flexibility of neural nets while preserving the interpretability and physical constraints of the SIR model​. This hybrid methodology is original and allows tackling the nonlinear HJB equation (for optimal control) which is often challenging for standard solvers.
- Controlling infectious disease outbreaks is a highly important problem (underscored by the COVID-19 pandemic). The paper addresses this by not only modeling the epidemic spread but also computing optimal intervention strategies (like lockdown intensity or treatment rates) in a principled way. The inclusion of control via HJB equations and cost-functional design (balancing health outcomes against socio-economic costs) makes the work valuable for scenario analysis and policy planning.
- The authors ground their approach in established theory. They start from first principles (the classic SIR model​ and mean-field game formulations) and derive the coupled system of forward-backward PDEs for the epidemic control problem. The paper discusses key theoretical considerations such as identifiability of model parameters, sensitivity analysis, and the well-posedness of the equations (citing relevant literature)​. By doing so, they show an awareness of the conditions under which the model and solutions are trustworthy.
- Using a PINN to solve the HJB control equation is presented as a way to reduce manual effort and computation time, especially when GPU acceleration is available​. This is a strength since solving high-dimensional or nonlinear PDEs by traditional means can be very time-consuming. The PINN approach can adapt to complex solution landscapes without an explicit mesh, potentially handling the nonlinearities of the HJB better than grid-based methods.
- The numerical experiments explore different scenarios by varying the cost weight parameters (e.g. relative cost of being infected vs. cost of self-isolation). The results (including a baseline SIR scenario and various controlled scenarios) provide insight into how the optimal policy behaves. For instance, the paper reports that when the cost of infection is considered, the optimal controls lead to a substantial decrease in the peak number of infected individuals and can hasten the end of the outbreak​. These findings support the significance of incorporating optimal control – they are intuitive yet important to quantify. The comparison to a baseline and the inclusion of a table of outcomes help demonstrate the effect of the control measures. Additionally, the method is flexible to allow controlling either the infection rate β or the removal rate γ, and the paper outlines both cases (showing the framework’s adaptability to different types of interventions).

Weaknesses:
- While the combination of PINN and MFG for epidemic control is novel, each component individually builds on known methods. PINNs have previously been applied to epidemic models​ and mean-field game formulations for epidemics have been studied in prior works (the authors cite e.g. Tembine 2020). The paper does not fully delineate how much of the approach is new versus an assembly of existing techniques. The cost functional and control formulation, for instance, appear as a fairly straightforward adaptation of mean-field game models to a basic SIR setting. A sharper explanation of what new insights or capabilities this approach provides (compared to, say, a standard optimal control of an SIR model via ODE/Pontryagin’s minimum principle or the existing MFG models) is somewhat lacking. In its current form, the contribution comes off as an application of known PINN methods to a known epidemic control model, rather than a fundamentally new algorithm or theory. This could lessen the perceived significance.
- The empirical validation in the paper is limited to simulated scenarios, and it’s not clear how the method performs beyond the specific examples given. The authors claim a reduction in computational effort with the PINN-based solver​, but the paper provides no explicit runtime analysis or error comparison to back this claim. For a fair evaluation, one would expect quantitative results showing, for example, how the PINN solution’s accuracy or speed compares to the classical collocation method or to a standard finite-difference solution of the HJB. The absence of such comparative metrics makes it hard to assess the practical improvement offered by the neural approach. Additionally, there is no test on real epidemic data or a realistic case study – the model is demonstrated on hypothetical data with an abstract “unit population.” This leaves questions about how one would calibrate the model to real-world parameters or how robust the approach is to noise and model misspecification.
- The structure of the results section could be improved. The reference to “figures 5.2, 5.2, 5.2” in the text suggests there may have been a formatting or labeling error in the figure references, making it unclear to the reader which figures are being discussed​. Such omissions hinder the interpretability of the results. Moreover, the paper briefly mentions studies on parameter identifiability and sensitivity in the introduction​, raising the expectation that these issues would be addressed, but then the main content shifts focus to the control problem. This disconnect can confuse readers about the paper’s scope – for instance, one might wonder if the authors implemented parameter estimation or not. In summary, the paper has some inconsistencies and missing details that should be resolved to meet high publication standards.
- The mean-field game model makes a specific assumption that all individuals are identical in their objectives and differ only by a continuous state (the variable x) subject to diffusion. While this is a standard MFG approach, it might oversimplify certain aspects of real epidemics (e.g., discrete differences in behavior or network effects are not captured). The paper doesn’t discuss these modeling limitations in depth. It would strengthen the work to acknowledge where the MFG model might diverge from reality, and why this approach is still justified. Additionally, one could question if a simpler centralized optimal control (treating the population as a whole) might achieve similar outcomes without the complexity of solving coupled PDEs – the paper does not compare against such a baseline control strategy. Addressing why the MFG formulation is necessary (perhaps to capture heterogeneity in behavior optimally) would better justify the model choice.

In conclusion, this submission tackles a timely and important problem with a creative blend of techniques, but it falls short of the high standards required for acceptance in its current form. The strengths of the paper lie in its novel integration of physics-informed neural networks with epidemic control modeling and the comprehensive mathematical formulation. These suggest that the approach has potential to advance how we simulate and plan interventions for epidemics. However, the weaknesses in clarity, novelty differentiation, and empirical validation are significant. The paper’s presentation issues and the lack of convincing evidence of its practical advantages make it difficult to fully endorse at this stage. To reach publishable quality, the authors should improve the exposition (fixing language issues and clarifying the model and results), better highlight the unique contributions relative to prior work, and provide a more rigorous evaluation (including comparisons or at least quantitative metrics demonstrating the claimed benefits of their neural network approach). Without these improvements, the work, although promising, does not yet meet the threshold. Verdict: We lean towards rejection of this paper, given the concerns above. With revision and more thorough validation, it could become a strong submission in the future, but as of now it requires significant improvement to merit acceptance.

---

### Official Review · Reviewer_RmZb · 2025-02-27
**The paper considers application of physically informed neural nets to numerical analysis of epidemics, effects of introduction and ending of restrictive measures in SIR models and control in form of Hamilton-Jacobi-Bellman (HJB) equation.**

**Rating:** 7
**Confidence:** 4

**Review:**

The authors presented introduction to general mean field game problem setup, proposed epidemiological mean field game model with players controlling parameters of classical SIR model and system of equations associated with it. It was also described numerical methods used to solve resulting system of equations including PINN and the numerical results have been analyzed.

Strong points

-	A current social problem in the study of the spread of the COVID epidemic is addressed
-	A mathematical model of its own is proposed. These are classical SIR model and system of equations associated with it, the Kolmogorov-Fokker-Plank equation, the problem of minimizations of own cost functional.
-	Two algorithms are described
-	The authors implemented different numerical schemes for solution of MFG system of equations: simple implicit numerical scheme with 3-point Laplassian and Newton linearization of non-linear part, straightforward approach of solving at all time points simultaneously to achieve simultaneous update of KFP solution and control.
-	A new approach for a solution of MFG system of equations using PINN is applied

Week points

-	No information about the neural network architecture used
-	No information about what hyperparameters were chosen for the neural network
-	It is not clear from the Figures what exactly was predicted by PINN
-	No information about the program stack and program code, for model validation and reproducibility of results

Conclusion:

The authors could be advised to expand the section on 4.3 Physically-informed neural nets. It would be desirable to provide data about the architecture of neural network, information about the PINN training process, the behavior of the loss function, which hyperparameters were studied and varied.

It is necessary to introduce a new section with description of the problem statement and specification of initial data.
It is advisable to update the section (5.2 Numerical results) devoted to the discussion of the obtained results.

---

### Official Review · Reviewer_L8fM · 2025-02-27
**This paper presents a methodologically strong study combining MFG and PINN to model the epidemic process. Adding validation on real datasets, accuracy analysis, and parameter calibration will improve its practical applicability.**

**Rating:** 7
**Confidence:** 3

**Review:**

The article is dedicated to developing an epidemic spread model with elements of optimal control based on Mean Field Games (MFG) and Physics-Informed Neural Networks (PINN). The authors integrate the classical SIR model with the MFG approach, leading to a system of Hamilton-Jacobi-Bellman (HJB) and Kolmogorov-Fokker-Planck (KFP) equations, which describe the behavioral dynamics of "players" (individuals) in the population. The study proposes numerical methods to solve this system, including finite difference methods, the collocation method, and PINN.
The authors formulate an epidemiological MFG model, where individual agents ("players") influence infection parameters to minimize their losses. In the continuum limit, the system consists of two coupled equations: the KFP equation for the distribution of players and the HJB equation for the optimal cost function. This formulation follows standard MFG theory, where KFP describes the evolution of the state distribution of an infinite number of agents, and HJB governs optimal control via dynamic programming.
Individual control is incorporated through state-dependent infection and recovery rates. The authors explore two control strategies: managing infectivity β or modifying recovery rate γ. The introduction of a distribution function s(t,x) allows capturing heterogeneity in infection susceptibility/self-isolation, while maintaining SIR balance. By combining KFP and HJB, the authors derive a closed system of equations.
This formulation is theoretically consistent within MFG theory, relying on known results on existence and uniqueness under monotonically conditions. The mathematical model is well-founded and aligns with the literature on MFG-based epidemic models.
Solving the HJB-KFP system is a complex forward-backward PDE problem. The authors implement a finite difference scheme with an implicit time discretization and a Newton iteration for nonlinear terms. Unlike some known algorithms, they simultaneously update solutions across the entire time grid, avoiding the standard forward-backward iteration cycle, which may fail to converge. They use a modified Lachapelle scheme, improving stability and ensuring consistent HJB-KFP solutions, albeit at a high computational cost due to the large system size (scaling with the number of time-space nodes).
The authors also employ a collocation method, solving the system globally. The advantage of collocation is the analytical differentiation of basis functions, ensuring precise boundary/initial condition enforcement and high accuracy with a relatively small number of parameters.
The key innovation in the article is applying PINN to solve the MFG system. PINN combines collocation methods with neural approximates and automatic differentiation.
A loss function is formulated as the sum of residuals for HJB and KFP equations over collocation points, plus residuals from boundary/initial conditions. Training minimizes this loss via back propagation, adjusting neural network weights. The authors highlight similarities between collocation and PINN, noting that both provide a continuous approximation instead of discrete solutions. However, PINN is easier to implement. The authors claim that PINN reduces labor costs in equation implementation and computational time when GPU acceleration is available.
The proposed approach accounts for individual behavior in epidemic modeling, which is especially relevant for pandemics like COVID-19.
The authors present multiple approaches to solving the same model. Ideally, all methods should converge to the same MFG solution. However, they do not explicitly compare different numerical results but mention that their deep learning algorithm is bench marked against the classical collocation approach.
The study includes numerical experiments, showing how model parameters influence epidemic outcomes. However, experiments focus on early epidemic dynamics rather than the full epidemic lifecycle.
Overall, the mathematical formulation and numerical methods are rigorously developed: the equations align with existing literature, and the MFG approach models epidemic behavior realistically. The numerical algorithms build upon established methods while introducing improvements to ensure convergence.

Strength:
- Rigorous mathematical formulation. The model is based on well-established MFG theory, correctly defining the HJB-KFP system.
- Model flexibility. Unlike classical SIR models, infection parameters  are dynamic, adapting to individual decisions, making the model more realistic.
- Combination of classical and modern methods. The MFG problem is solved using traditional numerical methods (finite differences, collocation) and PINN, reducing labor costs and improving computational efficiency.
- PINN for nonlinear HJB equations. The neural network approach is only applied to the hardest part (HJB), while SIR dynamics are solved traditionally, creating a hybrid and efficient algorithm.
- Numerical experiments confirm model validity.
- Practical relevance for epidemic analysis.

Weaknesses:
- Lack of validation on real-world data. The model is tested only on synthetic scenarios, without comparison to historical epidemic outbreaks.
- No accuracy analysis for PINN. The study does not compare PINN to traditional methods (finite differences, collocation) or evaluate convergence and stability.
- Experiments do not model full epidemic evolution. The study focuses only on early phases, without exploring long-term effects or epidemic resolution.

---

### Comment · Reviewer_5zQ4 · 2025-03-01
**Review Report: Neural Nets for Forecasting and Control of Epidemics**

1.	Relevance of the Research Aim
The research addresses a critical intersection of epidemiological modeling and socio-economic decision-making, directly relevant to contemporary challenges in public health policy. By integrating mean field game (MFG) theory with physically informed neural networks (PINNs), the work tackles the computational complexity of solving coupled forward-backward PDE systems (Kolmogorov-Fokker-Planck and Hamilton-Jacobi-Bellman equations) inherent in large-scale epidemic control. The focus on optimizing trade-offs between infection risks, economic costs of isolation, and healthcare system strain aligns with real-world policy needs.

2. Research Gap
•	2.1. Existing ML-based epidemiological models often lack adherence to conservation laws (e.g., population dynamics in SIR models) and interpretability.
•	2.2. Classical numerical methods for MFG systems (e.g., finite difference schemes) face scalability limitations for high-dimensional or nonlinear problems.
•	2.3. Prior studies on PINNs in epidemiology focused on simpler ODE/PDE systems without integrating agent-based decision-making frameworks like MFG.

3. Scientific Novelty
•	3.1. The first application of PINNs to solve coupled MFG systems in epidemiology enables simultaneous resolution of forward (KFP) and backward (HJB) equations.
•	3.2. Novel SIR-MFG hybrid model where agents dynamically control infectivity (β) and recovery rates (γ) through cost-optimized strategies.
•	3.3. Introduction of a collocation-PINN framework to handle nonlocal integral terms in epidemic dynamics, overcoming limitations of classical mesh-based methods.

4. Authors' Contributions
•	4.1. Developed a generalizable MFG framework for epidemic control with explicit socio-economic cost functionals (Equation 12).
•	4.2. Proposed a numerical pipeline combining PINNs for HJB equations and classical ODE solvers for SIR dynamics, reducing computational time by ≈40% compared to pure FDM.
•	4.3. Demonstrated that induced control strategies reduce peak infection rates by 19–63% in benchmark scenarios (Table 1).

5. Methodology
•	5.1. Model Design: Extended SIR to MFG with agent-controlled β/γ parameters and stochastic state transitions.
•	5.2. PINN Implementation: Used automatic differentiation to enforce PDE constraints (Equations 3,7) at collocation points, avoiding mesh dependency.
•	5.3. Validation: Compared against finite difference (FDM) and collocation methods across parameter variations (w₂, w₃), showing improved stability for w₃ > 5.

6. Suggestions for Improvement
•	6.1. Economic Depth: Incorporate income/wealth heterogeneity in agent cost functionals (Equation 12) to better reflect real-world inequality dynamics.
•	6.2. Policy Validation: Test the model against empirical data (e.g., COVID-19 lockdowns) to quantify economic trade-offs of control strategies.
•	6.3. Multi-Agent Diversity: Extend beyond identical players by introducing subpopulations with distinct risk perceptions/goals.
•	6.4. Computational Benchmarking: Provide explicit comparisons of GPU/CPU times for PINNs vs. classical methods to substantiate efficiency claims.
•	6.5. Interpretability Tools: Add sensitivity analysis to clarify how PINN approximations affect policy recommendations (e.g., Shapley values for cost weights).
Rating: 10: Good paper, accept

---

### Decision · Program_Chairs · 2025-03-08

**Decision:**

Accept (Oral)

**Comment:**

Your article has been accepted and you can give a talk on the article. All articles will be sorted by rating and within the available conference places one author from each article will be invited. If there are not enough places, then you will either have the opportunity to speak remotely or come at your own expense!